# Efficacy of Creatine Supplementation Combined with Resistance Training on Muscle Strength and Muscle Mass in Older Females: A Systematic Review and Meta-Analysis

**DOI:** 10.3390/nu13113757

**Published:** 2021-10-24

**Authors:** Ellem Eduarda Pinheiro dos Santos, Rodrigo Cappato de Araújo, Darren G. Candow, Scott C. Forbes, Jaddy Antunes Guijo, Carla Caroliny de Almeida Santana, Wagner Luiz do Prado, João Paulo Botero

**Affiliations:** 1Post-Graduate Program in Human Movement Sciences and Rehabilitation, Federal University of São Paulo (UNIFESP) Campus Baixada Santista, Santos 11015-020, Brazil; ellem.pinheiro@unifesp.br (E.E.P.d.S.); jaddy.antunes@unifesp.br (J.A.G.); 2Department of Physical Therapy, University of Pernambuco, Petrolina 56328-900, Brazil; rodrigo.cappato@upe.br; 3Faculty of Kinesiology and Health Studies, University of Regina, Regina, SK S4SOA2, Canada; darren.candow@uregina.ca; 4Department of Physical Education Studies, Brandon University, Brandon, MB R7A6A9, Canada; 5Agricultural College Dom Agostinho Ikas, Rural Federal University of Pernambuco-Recife, Recife 52171-900, Brazil; carlacaroliny@yahoo.com.br; 6Department of Kinesiology, California State University San Bernardino, San Bernardino, CA 92407, USA; wagner.prado@csusb.edu; 7Department of Human Movement Sciences and Rehabilitation, Federal University of São Paulo (UNIFESP) Campus Baixada Santista, Santos 11015-020, Brazil

**Keywords:** aging, sarcopenia, body composition, exercise, ergogenic aids, dietary supplements

## Abstract

Sarcopenia refers to the age-related loss of muscle strength and muscle mass, which is associated with a reduced quality of life, particularly in older females. Resistance training (RT) is well established to be an effective intervention to counter indices of sarcopenia. Accumulating research indicates that the addition of creatine supplementation (Cr) to RT augments gains in muscle strength and muscle mass, compared to RT alone. However, some evidence indicates that sex differences may alter the effectiveness of Cr. Therefore, we systematically reviewed randomized controlled trials (RCTs) investigating the efficacy of Cr + RT on measures of upper- and lower-body strength and muscle mass in older females. A systematic literature search was performed in nine electronic databases. Ten RCTs (*N* = 211 participants) were included the review. Overall, Cr significantly increased measures of upper-body strength (7 studies, *n* = 142, *p* = 0.04), with no effect on lower-body strength or measures of muscle mass. Sub-analyses revealed that both upper-body (4 studies, *n* = 97, *p* = 0.05) and lower-body strength (4 studies, *n* = 100, *p* = 0.03) were increased by Cr, compared to placebo in studies ≥ 24 weeks in duration. In conclusion, older females supplementing with Cr experience significant gains in muscle strength, especially when RT lasts for at least 24 weeks in duration. However, given the level of evidence, future high-quality studies are needed to confirm these findings.

## 1. Introduction

Musculoskeletal aging is associated with a progressive reduction in muscle strength (i.e., dynapenia) and muscle mass, which are hallmark characteristics of sarcopenia [1]. The reduction in muscle strength, which is the strongest predictor of health outcome measures in older adults [2], occurs more quickly than the reduction in muscle mass [3,4]. Furthermore, muscle weakness is highly associated with physical disabilities, chronic disease progression, and premature mortality [3]. Subsequently, the clinical importance of maintaining muscle strength is now considered the primary focus in offsetting sarcopenia, according to the European Working Group on Sarcopenia Research in Older People [2].

Considering the estimated longer life expectancy for females, there may be a greater likelihood of sarcopenia [5]. The Asian Working Group for Sarcopenia reported a prevalence of 8.6% in elderly females in community dwellings (>65 years) [6], while in the USA, 22.6% of older adults had sarcopenia [7]. The impact of sex on sarcopenia is controversial [8,9]; however, a high incidence of sarcopenia has been observed in females aged 60 years and over [5,10]. As such, effective interventions to prevent or reduce indices of sarcopenia in aging females are needed. The combination of creatine supplementation (Cr) [11,12,13,14] with resistance training (RT) (a well-established, safe, and effective intervention to increase/maintain muscle strength and muscle mass) [15,16] has shown promise for improving indices of muscle biology in older females. Previous systematic reviews and meta-analyses [17,18,19,20] indicate that Cr + RT increase measures of muscle strength and muscle mass in older males and females; however, sub-analysis on sex differences were not performed. There is evidence that sex related differences in response to Cr may exist. Females may have higher intramuscular creatine stores (at rest), which may blunt their responsiveness to exogenous Cr, thus they do not appear to experience reductions in muscle protein catabolism (compared to males) (for review see Smith-Ryan et al., 2021) [21]. Therefore, it is important to determine whether Cr + RT have significant muscle benefits in older females.

Considering the clinical significance of prevailing sarcopenia in older females, our purpose was twofold: (1) to systematically review the literature, and summarize the available scientific evidence on Cr + RT in older females; and (2), to perform meta-analyses on muscle strength and muscle mass to determine, with adequate statistical power and greater certainty, whether Cr + RT has anti-sarcopenic effects in older females.

## 2. Materials and Methods

### 2.1. Search Strategy, Study Selection, and Data Extraction

This review was conducted based on the Cochrane Manual for systematic reviews of interventions [22], and reported according to the Preferred Reporting Items for Systematic Reviews and Meta-Analyze (PRISMA-P) [23]. It was registered in the International prospective register of systematic reviews (PROSPERO), with registration number CRD42020221648. A systematic literature search was performed in Pubmed, EMBASE, Cochrane, Scopus, SportDiscus, CINAHL, Lilacs, Scielo, and Web of Science throughout May 2021. The searches included combinations of the following MESH terms: “Aged”, “Resistance training”, and “Creatine” and “Muscle strength”. No filters were applied, and we did not restrict the search by language or date of publication. Gray literature was also consulted. This step was carried out by E.S. and J.G. independently; when there were disagreements, J.B. was consulted for consensus. Figure 1 shows the flowchart of the study selection process. Studies that met the eligibility criteria were included. In addition to author and year of publication, the following characteristics of the study were extracted: details about the randomization process; blinding and allocation; sample size; ages; sex; inclusion and exclusion criteria; dose, frequency, duration of Cr and placebo (Pl) supplementation; and frequency and duration of the RT protocol. Whenever possible, we also collected compliance/adherence data relevant to the training and the supplementation protocol, as well as any adverse events that were reported. The primary outcome of this review was muscle strength, and we include muscle mass data when available. For each group and for each outcome, at both the pre- and post-intervention time points, sample sizes were included in the analyses, of which means and standard deviations were recorded. For those studies that did not report the necessary data for the analysis, the authors were contacted via e-mail.

### 2.2. Eligibility Criteria

Randomized controlled trials (RCTs) that met the following criteria were included: (1) examined the effects of Cr + RT on the muscle strength and muscle mass in older females (≥60 years) and/or postmenopausal females; and (2), studies with older adults ≥60 years of age, where the analysis of our main outcome (muscle strength) was reported (or made available) separately for females. Studies that used mixed creatine formula (that is, the inclusion of protein and other dietary supplements in creatine), examined other types of concurrent training, or involved subjects with neurodegenerative diseases were excluded.

### 2.3. Assessment of Methodological Quality of Included Studies

Included studies were evaluated by E.S. and R.A. independently for methodological quality (i.e., items 2–9) and statistical reports (i.e., items 10 and 11), using the Physiotherapy Evidence Database (PEDro) [24] scale of 0–10. Any disagreement between reviewers was resolved by consensus.

### 2.4. Data Analysis

Data extracted included the mean difference and the standard deviation between the pre- and post-training scores, reported in the included articles. In studies that presented only pre-and post-training values, we calculated the mean difference between post- and pre-training values. Standard deviation (SD) for change scores was estimated from the following equation, derived from the Cochrane Manual for Systematic Reviews of Interventions. [22]:SD change score = [(SDpre)2 + (SDpost)2 − 2 × (correlation between pre- and post-scores) × SDpre × SDpost]1/2

In this equation, we assume a value of 0.8 for the correlation between pre and post scores, as previously described by Chilibeck et al. [19].

Meta-analyses were performed in the Review Manager 5.3.4 software (The Cochrane Collaboration, Copenhagen, Denmark). For comparison between groups, the standardized mean differences (SMD) and their respective 95% confidence intervals were calculated. All analyses were performed using a random effects model.

The SMD used was the Hedges’ adjusted effect size. Meta-analyses were only conducted when the results of the studies were considered statistically homogeneous. To analyze the heterogeneity between studies, we used Chi-squared and I^2^ statistics, with I^2^ scores below 40% being considered insignificant [22].

To investigate the influence of intervention time, the studies were divided into subgroups of up to 14 weeks and ≥24 weeks of duration.

To assess muscle strength, the values recorded in multi-joint exercises of the upper-body (bench press or chest press) and lower body (leg press or hack squat) were analyzed. In the study by Johannsmeyer et al. [25], we considered the combined values obtained in leg press and hack squat to assess the muscle strength of the lower body. Effect sizes were interpreted according to the Hopkin scale (Hopkins, nd), with values < 0.20 indicating trivial, 0.20–0.59 indicating small, 0.60–1.19 indicating moderate, and values ≥ 1.20 indicating large effects. For all tests, we considered a significance level of 5%.

### 2.5. Analysis of the Level of Evidence

The GRADE system (Grading of Recommendations, Assessment, Development and Evaluation) was used to summarize the quality of the evidence and its strength of recommendation. GRADE provides four recommendation levels ranging from high to very low. As this is a review with experimental studies, the evidence came from the high level, with the possibility of downgrades according to five aspects: (1) risk of bias (mean PEDro score < 5); (2) inconsistency (I^2^ > 50%); (3) indirectness (when more than 25% of the studies used non-standard measures of muscle strength or muscle mass); (4) imprecision (grouping < 300 participants); and (5), publication bias or lack of its evaluation due to the small number of studies (*n* < 10) [26].

## 3. Results

After reading the full texts, ten articles met the eligibility criteria (Figure 1), and were included in the qualitative assessment. Of these studies, two were excluded from the meta-analyses, as, after contacting the authors, one reported that they no longer had the data [11], and the other did not respond [12]. Thus, eight studies were included in the meta-analysis of the main outcome [13,14,25,27,28,29,30,31], the characteristics of which are shown in Table 1. Of the articles that involved older adults, only data referring to females were extracted or requested (*n* = 3). Study participants (*n* = 176) were randomized to receive Cr or Pl during RT, which was performed two to three times/week. Mean age ranged from 56 to 70 years, and studies included healthy and postmenopausal females (*n* = 6); in postmenopausal females with osteopenia or osteoporosis (*n* = 1) or with knee osteoarthritis (*n* = 1) were included. The duration of the studies ranged from 12–52 weeks. Some studies adopted a Cr loading phase at a dosage of 20 g/day for 5–7 days, followed by 5 g/day (*n* = 3); others used a dosage of 5 g/day (*n* = 2), or a relative dosing strategy of 0.1 g/kg/day (*n*= 3). Six studies reported significant effects of Cr on muscle strength and muscle mass, while two found no effect. Only two studies reported adverse events (gastrointestinal discomfort or cramps), but none of these adverse events resulted in a loss at follow-up. The longest study (52 weeks) evaluated kidney and liver function and found no adverse effects (compared to Pl) from Cr.

Table 2 shows the individual score on each item, and the total scores of the studies on the PEDro scale. The quality ranged from 6 to 9, and the median PEDro score was 7. The randomization procedure was properly described in all of the included studies; however, only 40% adequately reported allocation procedures. Eight studies (80%) did not report whether the data analysis followed the intention-to-treat, and five (50%) of the studies showed a dropout greater than 15%.

Figure 2 shows the comparison of lower-body strength between groups that supplemented with Cr or Pl. Overall, there was no significant effect from Cr on measures of lower-body strength (*p* = 0.18; I^2^ = 0%; SMD = 0.22 [95% CI −0.10–0.55]). Sub-analysis performed on study duration showed no effect from Cr if the study lasted ≤ 14 weeks (*p* = 0.49; I^2^ = 0%; SMD = −0.20 [95% CI −0.75–0.36]). However, Cr significantly increased lower-body strength compared to Pl if the study lasted ≥ 24 weeks (*p* = 0.03; I^2^ = 0%; SMD = 0.44 [95% CI 0.04–0.84]). Collectively, the quality of evidence according to GRADE was rated as low (Table 3).

Quality of evidence:

High: We are very confident that the true effect lies close to that of the estimate of the effect.

Moderate: We are moderately confident in the effect estimate; the true effect is likely to be close to the estimate of the effect, but there is a possibility that it is substantially different.

Low: Our confidence in the effect estimate is limited; the true effect may be substantially different from the estimate of the effect.

Very low: We have very little confidence in the effect estimate; the true effect is likely to be substantially different from the estimate of the effect.

Reason for downgrade: Few studies with wide confidence interval and sample size lower than 300 (extremely serious imprecision).

Figure 3 shows the comparison of upper-body strength between groups that supplemented with Cr or Pl. Overall, Cr resulted in a significant increase in upper-body strength (*p* = 0.04; I^2^ = 0%; SMD = 0.35 [95% CI 0.02–0.69]). Similar to lower-body strength, sub-analysis performed on study duration showed no effect on upper-body strength from Cr if the study lasted ≤ 14 weeks (*p* = 0.44; I^2^ = 0%; SMD = 0.23 [95% CI −0.36–0.82]). However, Cr significantly increased upper-body strength compared to Pl if the study lasted ≥ 24 weeks (*p* = 0.05; I^2^ = 0%; SMD = 0.41 [95% CI 0.01–0.82]). The overall quality of evidence according to GRADE was rated as low (Table 3).

Cr, independent of study duration, had no effect on measures of muscle mass (Figure 4). The overall quality of evidence according to GRADE was rated as low (Table 3).

## 4. Discussion

The primary purpose of this meta-analysis review was to systematically evaluate the effects of Cr + RT on measures of muscle strength and muscle mass in older females. We chose to focus only on older females, since there is some evidence that females, in comparison to males, may have higher intramuscular creatine stores (pre-supplementation), which may blunt their responsiveness to exogenous Cr, and that Cr has no effect on indicators of muscle protein catabolism when RT is 12 weeks in duration (for review, see Smith-Ryan et al., 2021) [21]. Therefore, longer Cr periods may be required to produce significant muscle benefits in older females. Overall, Cr (independent of duration) had favorable effects on measures of upper-body strength, compared to Pl. These results support previous meta-analyses involving a combination of older males and females [17,18,19,20]. Sub-analyses revealed a unique finding in those studies lasting ≥ 24 weeks in duration: Cr increased upper- and lower-body strength compared to Pl. These muscle strength benefits from Cr may be clinically relevant, as greater upper-body strength is associated with an improved ability to perform activities of daily living (i.e., carrying groceries, lifting objects, household chores), and an overall higher quality of life in older females [32]. In addition, greater lower-body strength (the region most negatively affected by the aging process) [33] is associated with improved mobility [34], the ability to perform activities of daily living (i.e., climbing stairs, standing from a chair, household chores) [35], and a reduction in the risk of falls and subsequent fractures [36,37]. Furthermore, improvements in muscle strength reduce the incidence of chronic disease and premature mortality in older adults [3]. From an applied perspective, these findings may have applications for the design of effective Cr strategies for older females to improve muscle strength.

While no mechanisms were determined in this review and meta-analysis, greater muscle strength from Cr may be related to Cr’s ability to influence intramuscular phosphocreatine stores and actin-myosin cross-bridge cycling [13]. Furthermore, older adults have reduced phosphocreatine stores (compared to younger adults) in lower-body muscle groups (i.e., vastus lateralis) [19] that may stimulate a greater force generating response from Cr when these muscle groups are recruited (i.e., when performing squat, leg press, or knee extension exercises). Albeit speculative, Cr supplementation elevates intramuscular creatine stores, and increases exercise capacity which, over time, will translate to greater gains in muscle strength in older females. It is also possible that Cr increases calcium cycling into the sarcoplasmic reticulum, which would result in faster detachment of the actin-myosin cross-bridge, and subsequently increase force-generating capacity [38]. Unfortunately, no measure of calcium cycling was made, or muscle biopsies performed in the reviewed studies, which negates a definitive mechanistic conclusion.

Our lack of significant finding regarding muscle accretion from Cr contrasts with previous meta-analyses [17,18,19,20]. These contrasting findings occurred despite similar methodology of included studies, including supplement dose (5–20 g/day or 0.1 g/kg/day), training frequency (2–3 times per week), and duration of the intervention (12–52 weeks). Chilibeck et al., 2017 included studies that co-ingested creatine with other ingredients (e.g., protein or conjugated linoleic acid) [19]. However, a unique aspect of our analysis, albeit limited by a smaller sample size than these previous meta-analyses, was that it only involved older females. While not statistically significant, close examination of all individual studies involved in our analysis showed that Cr + RT increased or attenuated the rate of muscle loss compared to Pl. Mechanistically, Cr may have had some favorable effects on muscle biology by influencing muscle protein kinetics, satellite cells, growth factors, myogenic transcription factors, protein kinases involved in the mTOR signaling pathway, or inflammatory processes and pathways; for reviews, see Candow et al., 2021; Chilibeck et al., 2017; Forbes et al., 2021 [18,19,39]. Additional research determining the mechanistic actions of Cr on a much larger population cohort of older females is needed to determine with greater certainty and probability whether Cr can augment muscle mass during a RT program.

### 4.1. Quality of Evidence

Based on the GRADE system and recommendation for the main outcomes, we judged the overall certainty of the evidence in this review as low, after downgrading the scores due to issues related to imprecision (i.e., total cumulative sample size lower than 300). Thus, the available evidence is limited by the number of studies, sample sizes, and confidence interval amplitude, preventing us from reaching robust conclusions about the effects of creatine supplementation combined with resistance training on muscle strength and muscle mass gains in older females.

### 4.2. Adverse Events

In support of previous reviews [17,18,19], most studies reported no adverse events, even when renal and liver function were assessed. Thus, it appears that Cr is safe and well-tolerated strategy in older adults [40], and can be safely used by postmenopausal females [41].

### 4.3. Limitations

A major limitation of the current review was that only eight studies were quantified in the meta-analysis [13,14,25,27,28,29,30,31], which limits external generalizability. Furthermore, even with a low risk of bias (median PEDro score of 7), four of these studies [14,28,29,30] had withdrawal rates above 15%, and six studies [13,14,25,27,28,29] did not report whether the data analysis followed intention-to-treat; therefore, caution is warranted. A strength and unique aspect of our review was the inclusion of GRADE recommendations, which were not conducted in the previous reviews.

## 5. Conclusions

In conclusion, Cr + RT in a small cohort of older females enhanced muscle strength when the duration was at least 24 weeks; however, there was no effect on muscle mass. Overall, the certainty of the evidence is low, given the limited sample size, which may contribute to the imprecision of the observed effect sizes. Therefore, more clinical trials are warranted. Future research is required to examine long-term clinical health outcomes in older females.

## Figures and Tables

**Figure 1 nutrients-13-03757-f001:**
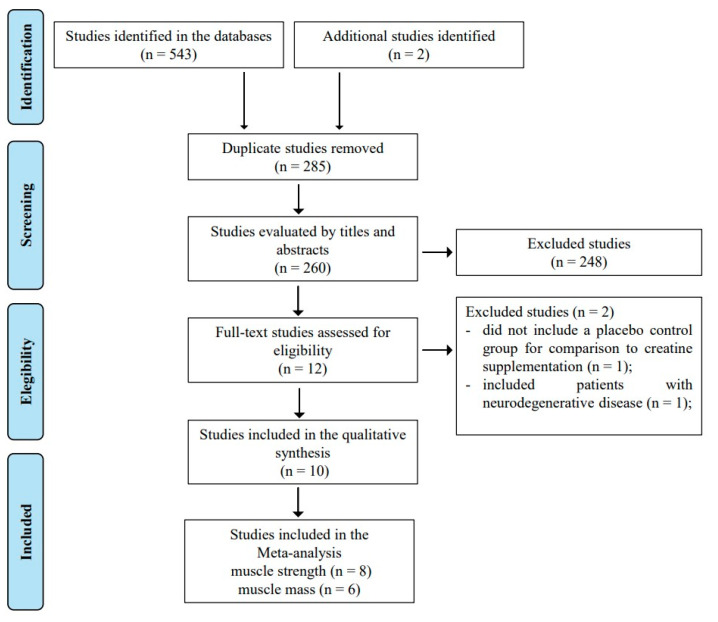
Flowchart of the study selection process.

**Figure 2 nutrients-13-03757-f002:**
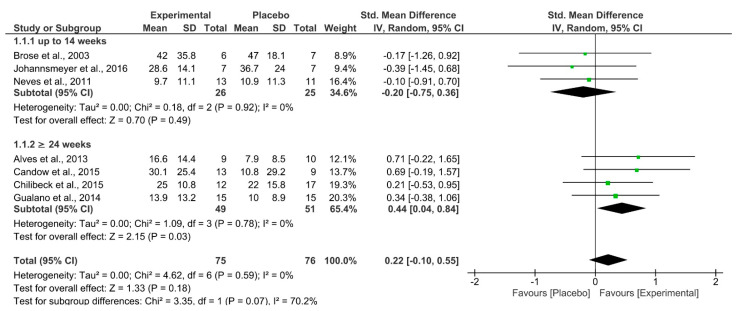
Forest plot for analysis of muscle strength of the lower body during the leg press or hack exercise, separated by subgroups according to study duration (up to 14 weeks and ≥24 weeks).

**Figure 3 nutrients-13-03757-f003:**
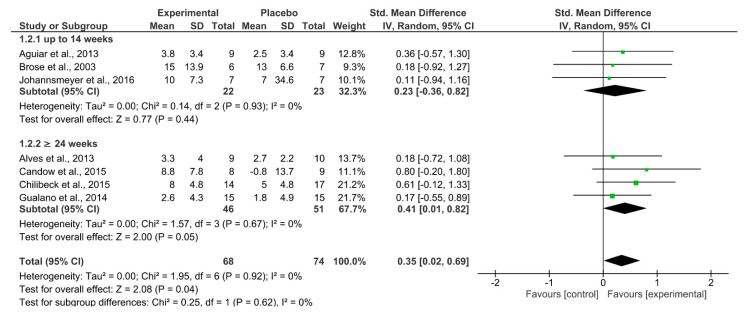
Forest plot for analysis of muscle strength of the upper body during the bench press or chest press exercise, separated by subgroups according to study duration (up to 14 weeks and ≥24 weeks).

**Figure 4 nutrients-13-03757-f004:**
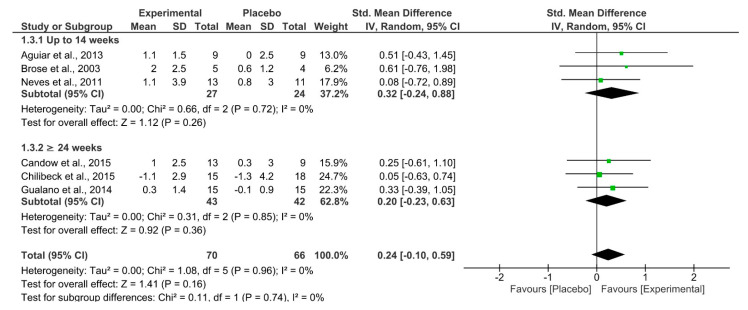
Forest plot for muscle mass, separated by subgroups according to study duration (up to 14 weeks and ≥24 weeks).

**Table 1 nutrients-13-03757-t001:** Characteristics of Included Studies.

Author (Year)	Population	*n =* Per Group	Age (year)	Intervention		Outcome Measure (*Italics Indicate Effect Significant Cr*)	Duration
				Supplementation	Training		
Aguiar et al. (2013) [27]	healthy women > 60 years	18 Cr = 9 Pl = 9	Cr: 64 ± 4 Pl: 65 ± 6	Cr or Pl: 5 g/day	RT 3 x/week	MS: 1RM (*bench press, knee extension* and *biceps curl*) BC: DEXA (*muscle mass*)	12 weeks
Alves et al. (2013) [28]	healthy older women > 60 years	22 Cr = 12 Pl = 10	Cr: 66.4 ± 5.6 Pl: 63.9 ± 3.8	Cr or Pl: 20 g/day for 5 days; followed by 5 g/day	RT 2 x/week	MS: 1RM (chest press and leg press)	24 weeks
Bermon et al. (1998) [11]	healthy older adults	8 Cr = 4 Pl = 4	Cr: 71.0 ± 1.9 Pl: 69.3 ± 0.4	Cr: 20 g Cr + 8 g glucose/day for 5 days; followed by 3 g Cr + 2 g glucose/day Pl: 28 g glucose/day for 5 days; followed by 5 g glucose/day	RT 3 x/week	MS: 1RM (leg press, chest press and leg extension)	52 days
Brose et al. * (2003) [13]	healthy older adults (women were postmenopausal)	13 Cr = 6 Pl = 7	Cr: 70.8 ± 6.1 Pl: 69.9 ± 5.6	Cr: 5 g Cr/day + 2 g dextrose/day Pl: 7 g dextrose/day	RT 3 x/week	MS: 1RM (leg press, chest press, arm flexion and Knee extension) BC: DEXA (*muscle mass*)	14 weeks
Candow et al. * (2015) [14]	healthy ≥ 50 years older adults (women were postmenopausal)	22 Cr = 13 Pl = 9	56 ± 5	Cr: 0.1 g Cr/kg/day Pl: 0.1 g malt/kg/day	RT 3 x/week	MS: 1RM (Cr before and after—*leg press* and *chest press*) BC: DEXA (Cr after—*muscle mass*)	32 weeks
Chilibeck et al. (2015) [29]	postmenopausal women	33 Cr = 15 Pl = 18	Cr: 57 ± 4 Pl: 57 ± 7	Cr: 0.1 g Cr/kg/day Pl: 0.1 g malt/kg/day	RT 3 x/week	MS: 1RM (*bench press* and hack squat) BC: DEXA (muscle mass)	52 weeks
Gualano et al. (2014) [30]	postmenopausal women with osteopenia or osteoporosis	30 Cr = 15 Pl = 15	Cr: 67.1 ± 5.6 Pl: 63.6 ± 3.6	Cr or Pl: 20 g/day for 5 days; followed by 5 g/day	RT 2 x/week	MS: 1RM (*bench press* and leg press) BC: DEXA (*muscle mass*)	24 weeks
Johannsmeyer et al. * (2016) [25]	Older adults (women were postmenopausal)	14 Cr = 7 Pl = 7	Cr: 59 ± 3 Pl: 58 ± 6	Cr: 0.1 g Cr + 0.1 g dextrose/kg/day Pl: 0.2 g dextrose/kg/day	RT 3 x/week	MS: 1RM (leg press, chest press, hack Squat and lateral pull-down) BC: DEXA (*muscle mass*)	12 weeks
Neves et al. (2011) [31]	postmenopausal women with knee osteoarthritis	24 Cr = 3 Pl = 11	Cr: 58 ± 3 Pl: 56 ± 3	Cr or Pl: 20 g/day for 7 days; followed by 5 g/day	RT 3 x/week	MS: 1RM (leg press) BC: DEXA (muscle mass, lower limb muscle mass)	12 weeks
Pinto et al. (2016) [12]	healthy older adults	27 Cr = 13 Pl = 14	Cr: 67.4 ± 4.7 Pl: 67.1 ± 6.3	Cr: 5 g Cr/day Pl: 5 g malt/day	RT 3 x/week	MS: 10RM (bench press and leg press) BC: DEXA (*muscle mass*)	12 weeks

* From the studies that included males and females, only data on females were extracted. Cr, creatine; Pl, placebo; malt, maltodextrin; RT, resistance training; MS, muscle strength; 1RM, 1—repetition maximum; 10RM, 10—repetition maximum; BC, body composition.

**Table 2 nutrients-13-03757-t002:** PEDro score (*n* = 10).

Study	Random Allocation	Concealed Allocation	Groups Similar at Baseline	Participant Blinding	Therapist Blinding	Examiner Blinding	<15% Dropouts	Intention to Treat Analysis	Between Group Difference Reported	Point Estimate and Variability Reported	Total (0–10)
Aguiar et al. (2013) [27]	Y	N	Y	Y	Y	Y	Y	N	Y	Y	8
Alves et al. (2013) [28]	Y	N	Y	Y	Y	Y	N	N	Y	Y	7
Bermon et al. (1998) [11]	Y	N	Y	Y	N	N	Y	N	Y	Y	6
Brose et al. (2003) [13]	Y	N	Y	Y	Y	N	Y	N	Y	Y	7
Candow et al. (2015) [14]	Y	Y	Y	Y	Y	Y	N	N	Y	Y	8
Chilibeck et al. (2015) [29]	Y	Y	Y	Y	Y	Y	N	Y	Y	Y	9
Gualano et al. (2014) [30]	Y	N	Y	Y	Y	Y	N	N	Y	Y	7
Johannsmeyer et al. (2016) [25]	Y	Y	Y	Y	Y	Y	Y	N	Y	Y	9
Neves et al. (2011) [31]	Y	N	Y	Y	Y	Y	Y	Y	Y	Y	9
Pinto et al. (2016) [12]	Y	N	Y	Y	Y	Y	N	N	Y	Y	7

Y: yes; N: no.

**Table 3 nutrients-13-03757-t003:** Summary of findings and quality of evidence (GRADE) for Cr.

No. of Studies	Study Design	Risk of Bias	Inconsistency	Indirectness	Imprecision	Outcome	Number of Subjects with Cr	Number of Subjects with Pl	Absolute Effect(95% CI)	Quality of Evidence (GRADE)	Importance
7	RCT	Not serious	Not serious	Not serious	Extremely serious ^a^	Upper-body Strength	68	74	SMD 0.35 (0.02 –0.69)	⨁⨁◯◯ LOW	Critical
7	RCT	Not serious	Not serious	Not serious	Extremely serious ^a^	Lower-body Strength	75	76	SMD 0.22 (−0.10–0.55)	⨁⨁◯◯ LOW	Critical
6	RCT	Not serious	Not serious	Not serious	Extremely serious ^a^	Muscle Mass	70	66	SMD 0.24 (−0.10–0.59)	⨁⨁◯◯ LOW	Important

CI, confidence interval; RCT, randomized clinical trial; SMD, standard mean difference; Cr, creatine supplementation; Pl, placebo. ⨁⨁◯◯ = low quality of evidence. ^a^ = wide confidence intervals and sample sizes lower than 300.

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
