# Peer review of "Efficacy of Creatine Supplementation Combined with Resistance Training on Muscle Strength and Muscle Mass in Older Females: A Systematic Review and Meta-Analysis"

_nutrients, 2021, doi:10.3390/nu13113757_

Round 1

Reviewer 1 Report

The manuscript provides a systematic review on the effects of resistance training and creatine supplementation on muscle parameters in elderly women. The review is well written, correctly conducted by the Cochrane manual and PRISMA, registered, presents flow chart with all the steps analyzed.

Author Response

Reviewer 1 Comments to Author:

The manuscript provides a systematic review on the effects of resistance training and creatine supplementation on muscle parameters in elderly women. The review is well written, correctly conducted by the Cochrane manual and PRISMA, registered, presents flow chart with all the steps analyzed.

Response: We thank the reviewer for their enthusiasm with our work.

Reviewer 2 Report

General comments

The purpose of this study was to systematically review randomized clinical trials investigating the efficacy of creatine + resistance training on measures of upper- and lower-body strength and muscle mass in older females. Creatine was found to significantly increase measures of upper-body strength with no effect on lower-body strength or measures of muscle mass. Sub-analyses revealed that both upper-body and lower-body strength were increased by creatine compared to placebo in studies lasting > 24 weeks in duration. While the focus on older women is original and of interest for the reader, there is not enough literature to be conclusive.

Major comments

- The number of RCTs included in the systematic review is rather low. In total, only 10 studies responded to the inclusion criteria for the systematic review and 8 for the meta-analysis. This tends to be in the lower acceptable range. In addition, some studies (for example Bermon et al and Brose et al) included few participants. The low number of studies together with the global low number of participants preclude from strong conclusions.

- This issue is amplified by separating the studies according to their duration (less than 14 weeks and more than 24 weeks), reducing the studies included in the meta-analysis to 3 or 4 studies.

- The discussion is very short and half of the discussion focuses on the possible mechanisms behind the effects observed in the present review, which is beyond the scope of the manuscript as it does not deal at all with the mechanisms of action of creatine. The paragraph about the putative mechanisms should be deleted and replaced by a discussion about the duration of the supplementation required to observe an effect in older women. This contrasts with what is known about creatine, i.e. after 2-3 months a plateau is obtained in intramuscular concentrations and the physiological effects tend to disappear. It is therefore quite surprising to see that 24 weeks supplementation induces larger effects than 14 weeks. This point should be discussed as this is a main outcome of the study.

Author Response

General comments: The purpose of this study was to systematically review randomized clinical trials investigating the efficacy of creatine + resistance training on measures of upper- and lower-body strength and muscle mass in older females. Creatine was found to significantly increase measures of upper-body strength with no effect on lowerbody strength or measures of muscle mass. Sub-analyses revealed that both upper-body and lower-body strength were increased by creatine compared to placebo in studies lasting > 24 weeks in duration. While the focus on older women is original and of interest for the reader, there is not enough literature to be conclusive.

            Response: We understand and agree with the reviewer’s comments. However, we still feel that this research is important (despite being limited). Unfortunately, research in general on females in the field of sport nutrition is limited. This meta-analysis, although limited is sufficient to detect a significant effect of creatine supplementation. This is important and provide guidance to a older females considering creatine to augment training. We have amended our conclusion to make it clear that more research is required.

Abstract:

“Given the level of evidence, future high-quality studies are needed to confirm these findings.”

Main document:

“Overall, the certainty of the evidence is low, given the limited sample size which may have contributed to the imprecision of the observed effect sizes.”

Major comments

- The number of RCTs included in the systematic review is rather low. In total, only 10 studies responded to the inclusion criteria for the systematic review and 8 for the meta-analysis. This tends to be in the lower acceptable range. In addition, some studies (for example Bermon et al and Brose et al) included few participants. The low number of studies together with the global low number of participants preclude from strong conclusions.

Response: We understand the point raised by the reviewer and have amended the conclusion (shown below). Furthermore, despite a relatively low number of studies, previous research in older adults (males & females) examining the impact of creatine + resistance included 199 participants for upper body strength and 213 for lower body strength and found significant benefits (Candow et al. 2014; Endocrine). As such, it is important to note that with a similar sample size, that are research found no effect of creatine on lower body strength and only a significant increase in upper body strength. These findings highlight the sex-based differences in response to creatine supplementation.

Conclusion:

“Overall, the certainty of the evidence is low, given the limited sample size which may have contributed to the imprecision of the observed effect sizes.”

- This issue is amplified by separating the studies according to their duration (less than 14 weeks and more than 24 weeks), reducing the studies included in the meta-analysis to 3 or 4 studies.

Response: Based on the reviewer’s comments, we have highlighted the low quality of evidence (GRADE result) mainly due to imprecision, as follows:

Discussion:

“Based on the GRADE system and recommendations for the main outcomes, we judged the overall certainty of the evidence for the additional effect of creatine use as low, after down grading the scores due to issues related to imprecision (i.e., total cumulative sample size lower than 300). The sample sizes of the individual trials are small, limiting the overall and sub-analyses. Thus, the available evidence is limited by the number of studies, sample size, and confidence interval amplitude, preventing us from reaching robust conclusions about the efficacy of creatine supplementation combined with resistance training for muscle strength and muscle mass gains in older females.”

- The discussion is very short and half of the discussion focuses on the possible mechanisms behind the effects observed in the present review, which is beyond the scope of the manuscript as it does not deal at all with the mechanisms of action of creatine. The paragraph about the putative mechanisms should be deleted and replaced by a discussion about the duration of the supplementation required to observe an effect in older women. This contrasts with what is known about creatine, i.e. after 2-3 months a plateau is obtained in intramuscular concentrations and the physiological effects tend to disappear. It is therefore quite surprising to see that 24 weeks supplementation induces larger effects than 14 weeks. This point should be discussed as this is a main outcome of the study.

Response: Thanks for your comment, we have shortened the sentences on mechanisms. However, we believe it is important to at least briefly mention how creatine works, particularly because we found a beneficial effect of creatine on muscle performance.

To answer why we found an increase in the longer duration studies whereas the shorter studies were less effective. We speculate that this is because creatine has a small effect each training session via enhance cellular energy capacity, which will induce small alterations in protein balance and over time lead to greater gains in muscle performance. In the present analysis it appears that studies < 14 weeks found no effect, whereas studies greater than 14 weeks (albeit the shortest study greater than 14 weeks was 24 weeks in duration) found improvements. We agree that intramuscular creatine stores can be saturated in a relative short time frame. For example, Hultman et al. (1992) found 20 grams/day for 5 days resulted in saturation and 3 grams/day for 28 days lead to a similar increase in creatine stores (~20% increase). An increase in intramuscular creatine stores increases exercise capacity, leading to an increase in exercise performance (i.e., more reps; Little et al. 2008 IJSNEM; PMID: 19033611), increase in training volume, and that leads to greater adaptations over time. The small increase in exercise capacity may take months to alter whole body changes, and our meta-analysis suggests that longer studies are required in older females to detect these improvements in muscular strength in older women. Lastly, Cr has no effect on indicators of muscle protein catabolism when RT is 12 weeks in duration in females (compared to males), which may also alter the study duration required to see a benefit compared to males.

Reviewer 3 Report

The manuscript entitled "Efficacy of creatine supplementation combined with resistance 2 training on muscle strength and muscle mass in older females: 3 A systematic review and meta-analysis" is well structured, the literature used is adequate to support what is elaborated in the text.

However, I have some suggestions for the authors:

1) Improve the quality of the figures

2) In the introduction, add some information on physical activity in general (doi: 10.3390 / ijerph17176065; doi: 10.3390 / antibiotics9060332; doi: 10.3390 / jcm9082540; doi: 10.3390 / antibiotics9060306; doi: 10.3390 / ijerph17249424) and on the use of creatine in women than in men (doi: 10.3390 / nu13030877; Hultman E, Bergstrom J, Spreit L, Soderlund K: Energy metabolism and fatigue. In Biochemistry of Exercise VII Edited by: Taylor A, Goll-nick PD, Green H. Human Kinetics : Champaign, IL; 1990: 73-92.; GL Close, DL Hamilton, A. Philp, LM Burke, JP Morton. New strategies in sport nutrition to increase exercise performance. Free Radical Biology and medicine, 98, 144-158; Hultman E, Soderlund K, Timmons JA, Cederblad G, Greenhaff PL: Muscle creatine loading in men.J Appl Physiol 1996, 81: 232-237; Willoughby DS, Rosene J: Effects of oral creatine and resistance training on myosin heavy chain expression .Med Sci Sports Exerc 2001, 33: 1674-81)

Author Response

- English language and style are fine/minor spell check required

Response: As solicited we have conducted a careful language edition in order to avoid spelling and grammatical issues.

1) Improve the quality of the figures

Response: We have improved the quality of the figures.

2) In the introduction, add some information on physical activity in general (doi: 10.3390 / ijerph17176065; doi: 10.3390 / antibiotics9060332; doi: 10.3390 / jcm9082540; doi: 10.3390 / antibiotics9060306; doi: 10.3390 / ijerph17249424) and on the use of creatine in women than in men (doi: 10.3390 / nu13030877; Hultman E, Bergstrom J, Spreit L, Soderlund K: Energy metabolism and fatigue. In Biochemistry of Exercise VII Edited by: Taylor A, Goll-nick PD, Green H. Human Kinetics : Champaign, IL; 1990: 73-92.; GL Close, DL Hamilton, A. Philp, LM Burke, JP Morton. New strategies in sport nutrition to increase exercise performance. Free Radical Biology and medicine, 98, 144-158; Hultman E, Soderlund K, Timmons JA, Cederblad G, Greenhaff PL: Muscle creatine loading in men.J Appl Physiol 1996, 81: 232-237; Willoughby DS, Rosene J: Effects of oral creatine and resistance training on myosin heavy chain expression .Med Sci Sports Exerc 2001, 33: 1674-81)

Response: We have carefully reviewed each of the suggested references. We have added the references that we felt would add to the quality of the manuscript.

Round 2

Reviewer 2 Report

The too low number of articles dealing with the research question does not allow any valuable interpretation and conclusion, certainly not for a systematic review and meta-analysis.

Author Response

The too low number of articles dealing with the research question does not allow any valuable interpretation and conclusion, certainly not for a systematic review and meta-analysis.

Response: We believe, despite the limited number of studies that this research is important. We agree that more research is warranted. However, similar sample sizes in meta-analyses that included males found that creatine was able to augment resistance training gains in muscle mass, and both upper and lower body strength. These contrasting findings highlights that there may be sex based differences. We have highlighted the limited number of studies/participants as a limitation.